# Copy Number Variation Analysis Revealed the Evolutionary Difference between Chinese Indigenous Pigs and Asian Wild Boars

**DOI:** 10.3390/genes14020472

**Published:** 2023-02-12

**Authors:** Shuhao Fan, Chengcheng Kong, Yige Chen, Xianrui Zheng, Ren Zhou, Xiaodong Zhang, Xudong Wu, Wei Zhang, Yueyun Ding, Zongjun Yin

**Affiliations:** 1College of Animal Science and Technology, Anhui Agricultural University, Hefei 230036, China; 2School of Pharmacy, Anhui University of Chinese Medicine, Hefei 230036, China; 3Key Laboratory of Pig Molecular Quantitative Genetics of Anhui Academy of Agricultural Sciences, Anhui Provincial Key Laboratory of Livestock and Poultry Product Safety Engineering, Institute of Animal Husbandry and Veterinary Medicine, Anhui Academy of Agricultural Sciences, Hefei 230031, China

**Keywords:** CNV, CNVR, Anqingliubai pig, wild boar, QTL

## Abstract

Copy number variation (CNV) has been widely used to study the evolution of different species. We first discovered different CNVs in 24 Anqingliubai pigs and 6 Asian wild boars using next-generation sequencing at the whole-genome level with 10× depth to understand the relationship between genetic evolution and production traits in wild boars and domestic pigs. A total of 97,489 CNVs were identified and divided into 10,429 copy number variation regions (CNVRs), occupying 32.06% of the porcine genome. Chromosome 1 had the most CNVRs, and chromosome 18 had the least. Ninety-six CNVRs were selected using VST 1% based on the signatures of all CNVRs, and sixty-five genes were identified in the selected regions. These genes were strongly correlated with traits distinguishing groups by enrichment in Gene Ontology and Kyoto Encyclopedia of Genes and Genomes pathways, such as growth (CD36), reproduction (CIT, RLN), detoxification (CYP3A29), and fatty acid metabolism (ELOVL6). The QTL overlapping regions were associated with meat traits, growth, and immunity, which was consistent with CNV analysis. Our findings increase the understanding of evolved genome structural variations between wild boars and domestic pigs, and provide new molecular biomarkers to guide breeding and the efficient use of available genetic resources.

## 1. Introduction

Sus scrofa provides approximately 50–60% of the meat needed in daily life. Wild boars have been an important source of meat since humans began raising them. Wild boars have many differences compared to domestic pigs, including carcass fatness, loin area, oxidative glycolysis, muscle fibers, and other meat qualities [1]. To better understand the role of CNV in the evolution of meat quality and other traits, high-throughput sequencing was used in this study to analyze genome differentiation between wild boars (Sus scrofa, SS) and domestic Anqingliubai pigs (AQ).

Whole-genome resequencing has recently become widely used in animal breeding, combining several mainstream analysis methods to explore differences between individuals and groups. As a commonly used structural variation analysis method in resequencing, it has long been neglected in domestication studies [2]. Copy number variation (CNV) is another valuable way to understand scientific and genetic breeding [3], defined as unbalanced variants of the genome structure at least 50 bp long and, at most, mega-bases long in comparison to the reference [4]. These variations can be divided into deletion (DEL) and duplication (DUP). As an important source of genetic variation, CNV plays an important role in phenotypic evolution, disease susceptibility, and environmental adaptation [5]. The results of reproductive performance have shown that CNV is significantly correlated with phenotypic differences, and the contribution of CNV to complex phenotypes is far greater than that of single-nucleotide mutations (83.6% vs. 17.7%) [6,7]; moreover, it has been widely applied in human diseases, as well as in the domestication of animals and plants [2,8,9]. In humans, the CDCA4 gene is highly expressed in tumors with poor overall and disease-free survival, and positively correlated with CNV [8]. Chao et al. [10] drew a genomic CNV landscape and identified that the RXFP2 gene was strongly selected, which is functionally related to the horn type in sheep. Another study of three different cattle cultivars concluded that 78 CNVRs were different and could be selected; the CNVs represented the global diversity of cattle populations influenced by history and geography [11]. Previous CNV studies reported in pigs were detected using SNP-array and next-generation sequencing methods. Zheng et al. [9] identified 12,668 CNVRs in Meishan and Duroc pigs by dividing them into three groups, measuring the length of the total CNVRs, and finding that the AHR gene could stimulate reproductive performance. A total of 1693 pigs from 18 populations were detected by Chen et al. [12] using Porcine SNP60 BeadChip and PennCNV algorithm. As a result, 1315 putative CNVs belonging to 565 CNVRs were identified, and among them, seven copy number variable genes were related to carcass length, backfat thickness, abdominal fat weight, length of scapular, intermuscle fat content of logissimus muscle, body weight at 240 day, glycolytic potential of logissimus muscle, mean corpuscular hemoglobin, mean corpuscular volume, and humerus diameter. Revilla et al. [13] identified 1279 CNVs and 540 CNVRs containing 245 genes using next-generation sequencing data and selected the relevant candidate genes involved in backfat, fatty acid composition, and growth-related traits. Wang et al. [14] also detected CNV-genes and CNV-miRNAs in Duroc using 50K SNP arrays; they found 39 CNV-miRNAs in addition to 1096 CNV-genes, and CNV-miRNAs had more target genes compared to non-CNV-miRNAs. The CNV-genes and miRNAs they focused on were involved in the lipid metabolism.

Anqingliubai pigs (AQ) have a long history in Anhui province; they are praised for their precociousness, easy fattening, thin fur, thin hair, high reproductive power, good meat quality, rough feeding resistance, and strong resistance to stress. Due to these advantages, they are the first choice for the first female parent of ternary hybrid pigs and one of the important stock breeds for high-quality pig breeding. Domestic pigs are domesticated from wild boars, but different artificial selections have a great impact on different signatures [15], similar to AQ and other local pigs. It is critical to protect the resources of indigenous pigs in Anhui, to determine their superior traits compared to wild boars, and to clarify the genetic diversity between domestic breeds and wild breeds, such as Asian wild boar. In our previous work, the SNP of AQ was identified to analyze the runs of homozygosity (ROH) [16]. In this study, we identified the CNVs of AQ and Asian wild boars (SS) using high-throughput resequencing to indicate the CNVs and CNVRs, then annotated the functions of genes overlapping with different CNVRs.

## 2. Materials and Methods

### 2.1. Sequencing Data Reference

In this study, we used 30 pigs produced in our previous work, including 24 AQ from Taihu county (Anqing city, Anhui province) and 6 Asian wild boars from Mengla county (Xishuangbanna Dai Autonomous Prefecture, Yunnan province). In our previous work, these samples were resequenced using the Novaseq6000 (Illumina Inc., San Diego, CA, USA) NGS platform at Genedenovo Biotechnology Co., Ltd. (Guangzhou, China). The average depth of these data was ten generations of porcine genome. The analytical procedures for resequencing were adapted from Wu et al. [16]. Based on sequencing data, 935.04 Gb of raw data of the population were obtained and submitted to the National Center for Biotechnology Information (NCBI) database under the accession number PRJNA699491.

### 2.2. CNV and CNVR Detection

We utilized CNVnator software (version 0.3.2) to detect copy variation [17]. As our previous work suggested, we referred to their methods to filter the high-confidence CNVs with the following standards: (1) e-value < 0.01; (2) RD of deletion region < 0.7; (3) RD of duplication region > 1.3; and (4) length of CNV > 1 kb. GCTA software (version 1.24) was used to perform principal component analysis (PCA), and Treebest software (version 1.92) was used to draw the evolutionary tree.

The results of autosome CNVs was merged with CNVRs using CNV_overlap (https://github.com/bjtrost/TCAG-WGS-CNV-workflow, accessed on 18 May 2022) [18]; the CNVs would be connected as long as there was 1 bp intersection between different samples.

### 2.3. CNVR Selection

The parameter VST was calculated to analyze the genetic distance between populations: VST = (Vtotal − (Vpop1 × Npop1 + Vpop2 × Npop2)/Ntotal)/Vtotal, where Vtotal is the total variance in the copy number between the two groups and V1 and V2 are variances in the copy numbers of population 1 and population 2, respectively. N1 and N2 are the numbers of samples of population 1 and population 2, respectively. Ntotal is the total number of samples. We compared the genetic distance between groups using the mean VST values [19,20].

### 2.4. Functional Annotation of CNVRs

KOBAS 3.0 is a web server for gene/protein functional annotation (annotation module) and functional gene-set enrichment (enrichment module). Thus, to provide insights into the functional enrichment of the CNVR-overlapping genes, we performed Gene Ontology (GO) and Kyoto Encyclopedia of Genes and Genomes (KEGG) pathway analyses for the genes in CNVRs using KOBAS 3.0 (http://kobas.cbi.pku.edu.cn/kobas3/?t=1, accessed on 18 May 2022) [21].

In addition, the pig QTL Database (AnimalQTLdb, https://www.animalgenome.org/cgi-bin/QTLdb/index, accessed on 18 May 2022) was used to annotate the genes in the CNVRs [22].

## 3. Results

### 3.1. CNV and CNVR Detection

Using the CNVnator software, 97,489 CNVs on 18 autosomes were detected, including 41,454 DUPs, 34,678 DELs, 9622 inter_DUP DELs, and 11,735 inter_DEL DUPs. The two species had different distribution patterns; the number of DUPs was larger than that of DELs in AQ, in contrast to the SS.

In summary, after integrating the overlapping CNVs, we identified 10,429 high-confidence autosomal CNVRs in the two populations for subsequent analysis, accounting for 7.69% of the length of the porcine genome, 7.66% in AQ, and 1.74% in SS (Appendix A). A total of 10,279 CNVRs were from the AQ population and 4479 were from SS populations, with sizes ranging from 600 to 767,000 bp in AQ pigs, and 600 to 193,500 in SS pigs. The median and average sizes of each autosome in the two populations were also shown in Appendix A, respectively. There were 4859 deletions (average 9804 bp) and 5570 duplications (average 28,273 bp) in the two populations. Most of them were identified in the AQ population (4731 deletions and 5548 duplications) compared to those in the SS (3109 deletions and 1370 duplications). The length statistics and coverage of each autosome in the two populations were shown in Appendix A.

The distribution of CNVRs was not proportional to the length of chromosomes in the porcine genome. Chromosome 1 had the highest number of CNVRs in both populations, AQ, and SS, with 1175, 1157, and 457 CNVRs, respectively. Chromosome 18 was the least abundant, with 176, 175, and 78 CNVRs, respectively. The number of CNVRs on chromosome 15 was less than that of chromosome 5 in both populations, but chromosome 15 was longer. We analyzed the distribution of CNVRs on each chromosome, including their numbers and locations. The results are shown in Table 1 and Figure 1. In total, the frequency of DUP was 1.15 times greater than DEL; AQ has the consistent result 1.17, but the opposite was observed in SS with result 0.44 (Appendix A).

Density was calculated as the total number divided by chromosome length. We found that the density of CNVRs on each chromosome and population varied. The average densities of both populations, AQ, and SS populations were 4.6, 4.54, and 1.98, respectively. This illustrates that AQ had more than twice as many CNVR events as SS. In both populations, chromosome 2 had the highest density (6.79) and chromosome 18 had the lowest density (3.14).

To further discover the correlation of chromosome length and CNVRs, Pearson correlation between chromosome length, CNVR quantity, and CNVR length was calculated. The correlation coefficient between chromosome and CNVR quantity was 0.91, and that between CNVR length was 0.79, indicating that chromosome length is more closely related to CNVR quantity.

### 3.2. PCA and Evolutionary Tree Analysis

PCA was used to distinguish reproducible differences between AQ and SS populations. Samples of the two varieties were separated on two principal components, with percentages of 10.56% and 5.84%, respectively (Figure 2).

Moreover, to verify the repeatability of the samples in the two populations, we calculated the genetic distances between all individual samples using an evolutionary tree (Figure 3). It was shown that samples from the same groups had a closer distance relationship and could be easily distinguished from another group. This analysis, which separates samples into clusters, supports the conclusions obtained by the PCA analysis.

### 3.3. Differential CNVRs Measurement

To further discover the differential CNVRs between the two groups, we referred to the parameter VST to filter all CNVRs. In summary, the 823 groups of differential CNVRs (Appendix A, Appendix A) were identified with VST over 0.11 (top ten percent) and 96 CNVRs (Appendix A) with VST over 0.55 (top one percent) between AQ and SS pigs (Figure 4).

### 3.4. Functional Analysis

To better understand the function of the group-differential CNVRs, we utilized the functional enrichment of the GO and KEGG databases to annotate genes overlapping with differential CNVRs. The functional genes that overlapped with the top 1% and 10% of CNVRs are shown in Figure 5 and Figure 6 (Appendix A). In the top 10% CNVRs, 1097 overlapping genes were enriched in 2980 GO terms, and 65 genes were enriched in 804 GO terms in the top 1% CNVRs. However, most genes were enriched in level 1 GO terms, such as cellular component (GO:0005575), molecular function (GO:0003674), and biological process (GO:0008150). In the level 2 GO enrichment, only 38 GO terms were enriched in 20 biological processes, 7 molecular functions, and 11 cellular components in the top 1% group-differential CNVRs. These GO terms included growth (GO:0040007), development (GO:0032502), reproduction (GO:0000003 and GO:0022414), and immune system processes (GO:0002376), among others.

In KEGG pathway analysis, 219 and 57 pathways were enriched in the top 10% and top 1% of CNVRs, respectively. The TNF signaling pathway (ko04668), apoptosis (ko04210), fatty acid elongation, metabolism (ko00062 and ko01212), autophagy (ko04140), and JAK-STAT signaling pathway (ko04630) were enriched in the top 1% group of differential CNVRs.

To illustrate the regulatory relationships, interaction networks between genes associated with the group-differential CNVRs were constructed.

### 3.5. QTLs Overlapping with Identified CNVRs

To assess genetic effects, we compared the detected CNVRs with previous research on porcine QTLs in the AnimalQTLdb. There were 1591 group-differential CNVRs in the top 10% of VST associated with the QTLs and 208 CNVRs in the top 1% (Appendix A). These overlapping QTLs include average daily gain, meat color, traits 24 h post-mortem, and drip loss, which provide important information for pig molecular breeding.

## 4. Discussion

Based on our previous research, AQ pigs have good meat quality and adaptability [23]. As an excellent local breed, AQ pigs were domesticated from wild boars, the ancestors of domesticated pigs. In order to provide a unique opportunity for elucidating the genetics basis of domestication and further promote the breeding of pigs, copy number variations based on next-generation sequencing were selected to analyze the genetic difference between AQ and SS.

In recent years, the existence and importance of CNVs have received extensive attention in genomic research. As a source of genetic diversity, approximately 4.8–9.5% of the genome contributes to CNVs and affects the occurrence and progression of diseases, clinical applications, group evolution, and selection [20,24,25,26]. In humans and mice, CNVRs were not chosen positively, as we thought the salutary CNVRs were retained because of the elimination of detrimental CNVRs [27]. The emergence of CNVs is one of the driving factors of species evolution through gene dosage, gene disruption, gene fusion, and position effects.

Different methods and sequence depth affect CNVR numbers. In this study, 10,429 autosomal dominant CNVRs were identified, far more than the 348 CNVRs identified by Wang et al. [28] using SNP genotype arrays between Chinese pigs and European pigs. Though the amount has a large difference, the function of CNVR overlapping genes was similarly focused on disease resistance, reproduction, and muscle development. Compared to the study of Suhuai pigs with similar sequence depth, which were distributed in Jiangsu province and near AQ pigs [29], the number of CNVRs was 11,173, similar to our study. However, when compared with the study in which Zheng et al. [9] identified 12,668 CNVRs in Meishan pigs using a high-throughput sequence, the number of CNVRs identified in AQ and SS still has the potential to grow with deeper sequences and a new approach to analysis. Contrary to their results, the percentage of DUP was higher in our result, but the functions of CNVRs were enriched consistently, such as disease and immunity, reproduction, development, and sensory perception, which could possibly explain different evolutionary regulation in AQ pigs.

Using functional enrichment of the genes overlapping with group-differential CNVRs, many biological and molecular functions were identified, such as growth, development, reproduction, and immunity, which is consistent with previous studies. These genes provide a direction for studying the relationship between CNV and pig production. As is well known, domestic pigs have higher fertility than wild boars. We identified several genes involved in GO terms related to reproduction. Relaxin (RLN) is a pregnancy hormone in mammals and widely exists in multiple reproductive tissues of females, but only in sperm in males [30]. It is not only produced in reproductive organs to affect pregnancy and reproduction, but also in heart atria to promote the dilation of blood vessels in the heart to regulate cardiovascular disease, which could be one of the influencing factors of boars living in the low-oxygen environment of the plateau [30,31]. The RXFP family is involved in a range of reproductive-related functions such as follicle growth and ovulation [32]. The accumulation of RLN protein during oocyte maturation is very important and has a significant effect on subsequent reproductive processes. Citron Rho-interacting kinase 1, a tissue-specific ser/thr kinase encompassing the Rho-Rac-binding protein Citron, which can regulate the growth of human bladder cancer cells [33], was enriched in the GO terms “reproduction” and “reproductive process”. MNAT1, also known as NAT1, is an assembly factor of CDK-activating kinase (CAK), which can regulate maturation/M-phase-promoting factor (MPF) by phosphorylating its catalytic subunit [34]. It can affect normal meiotic progression during oocyte maturation, which may be one of the factors affecting reproduction between different varieties.

As shown in a previous study, CNVR-overlapping genes enriched for drug detoxification and immunity are highly significant in cattle [35]. Similar to past research, we identified the gene CD36 (platelet glycoprotein 4) in chromosome 9, which was enriched in immunity and growth, and was reported as a candidate gene for immunity traits in humans and pigs [36]. CD36 was enriched in phagosome, cholesterol metabolism, and especially fat digestion and absorption in KEGG pathway enrichment, existing on the surface of many immune cells and playing an important role in phagocytosis as a receptor [37]. In addition to being involved in phagocytosis, CD36 is also a fatty acid transporter that can bind to long-chain fatty acids [38,39]. Zhou et al. [40,41] found that the feeding of exogenous short-chain fatty acids could decrease the expression of CD36 because an increase in CD36 leads to obesity and type 2 diabetes. Li et al. [42] found that the addition of betaine regulated animal lipid metabolism by increasing the abundance of CD36 in finishing pigs. CYP3A29 (cytochrome P450) was also identified as another immunity-involved gene on chromosome 3, with the CNV of CYP450 genes often affecting their splicing and expression but not their transcription and structure [43]. The main function of CYP450 genes is to catalyze a large number of chemical reactions, such as the oxidative biotransformation of most drugs and lipophilic xenobiotics [43,44]. The CNV of CYP450 genes, including CYP3A29, may be a predicted role that helps animals adapt to dynamically changing environments, as reported in previous studies of swine CNVs [13,45]. Furthermore, the expression levels of CYP450 genes can affect androstenone levels in the Duroc and Landrace populations [46]. It was also found to be enriched in the lipid metabolism pathway in the KEGG database.

In addition to the CYP450 gene family, the UDP-glucuronosyltransferase gene family was also enriched in the drug metabolism–cytochrome P450 pathway, which is involved in a wide range of functions. In KEGG pathways, the UGT families participated in bile secretion, steroid hormone biosynthesis, pentose, and glucuronate interconversions. GO terms were enriched for metabolic processes, cell parts, membrane parts, response to stimulus, binding, and catalytic activity. As a study model, UGT and P450 act synergistically to metabolize drugs in human and porcine livers as excellent enzyme activators [47,48]. We found two members of the UGT family, UGT2C1 and UGT1A10, to be consistent with the findings of a previous study [45].

The olfactory receptor (OR) gene family has a large CNV in humans [49]. Although olfaction is unnecessary in humans, it plays a significant role in the survival of mammals by locating food, scent of mates, and natural enemies [50]. As the largest gene superfamily in swine, more than 1000 OR genes have been identified in the genome Sscrofa10.2 [51]. The common expansion of OR genes in swine suggests a unique olfactory sensory pathway in swine compared to other mammals. Owing to the different geographical locations between Anhui province and Tibet, the two populations may have different ways of smelling. The CNV of ORs detected in our data was OR7A10, revealing a difference in environmental adaptability between the two populations.

In addition to GO terms, genes enriched by the KEGG pathway also provided valuable information. Elongation of long-chain fatty acids (ELOVL6), which are enriched in fatty acid elongation, metabolism, and biosynthesis of unsaturated fatty acids, plays a key role in fatty acid composition in pigs. In two independent genome-wide association studies (GWAS), ELOVL6 and its family members were related to the content of fatty acids and other meat quality traits by regulating the activity of enzymes in hybrid breeds and several Chinese indigenous breeds [52,53]. The ELOVL6 gene has been reported to be the major cause of QTL for palmitic and palmitoleic acid content on pig chromosome 8, and the SNP of the two ELOVL6 isoforms could decrease its expression through promoter methylation [54]. We obtained new directions for the selection of excellent domestic pig breeds and the identification of biomarkers from our findings.

Although some traits related genes were reported in our research, there were still some limitations which should not be neglected. The sample size of our research (AQ, *n* = 20; SS, *n* = 6) was not as balanced and cannot completely represent the populations; it can also affect the VST calculation. In fact, the population of wild boars are much less abundant than indigenous pigs and are protected by the Wildlife Act. In contrast, the function of these reported genes was already annotated with databases such as GO and KEGG, and there is still a need for further studies to comprehend the genetic mechanism on traits. The limitations of our work should be improved in our next study. 

## 5. Conclusions

In this study, high-throughput resequencing was performed to identify CNVs and CNVRs to improve and supplement Chinese indigenous porcine CNV resources. Genome-wide distribution of CNV and CNVRs has been described, providing an important resource for future research. After identifying CNVs and CNVRs, 96 Vst top 1% CNVRs were selected and enriched by GO and KEGG. Then, we identified candidate genes overlapping with these CNVRs, which were related to the phenotype traits. To further evaluate the relationship between CNVRs and economic traits, we compared the selected CNVRs with QTLs in the animal QTL database. Our study provides new insights into genome structure variations in Chinese indigenous pigs and wild boars, helping to better understand the evolutionary relationship between wild boars and domestic pigs. In our study, additional molecular biomarkers were screened to provide evidence for future breeding.

## Figures and Tables

**Figure 1 genes-14-00472-f001:**
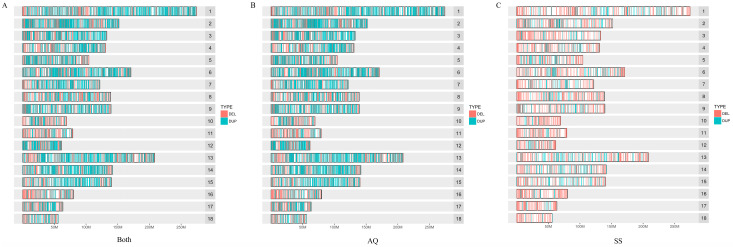
The distributions of CNVRs on 18 autosomal chromosomes in both populations (**A**) and each individual population; (**B**) represents AQ and (**C**) represents SS. The CNVRs were classified into two types: the red and light blue colors represent deletions and duplications, respectively.

**Figure 2 genes-14-00472-f002:**
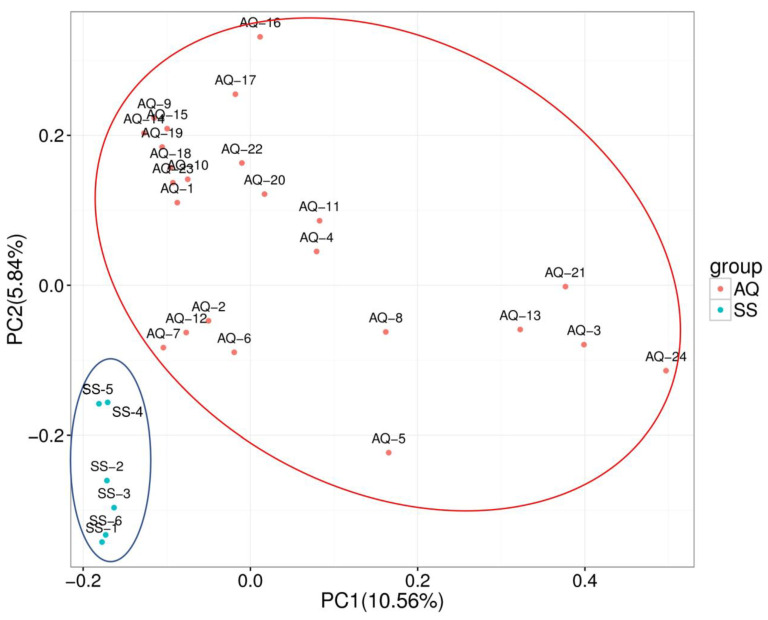
Principal component analysis (PCA) for the two main components of 30 individuals in two populations; pink nodes and light blue nodes represent the AQ and SS groups, respectively.

**Figure 3 genes-14-00472-f003:**
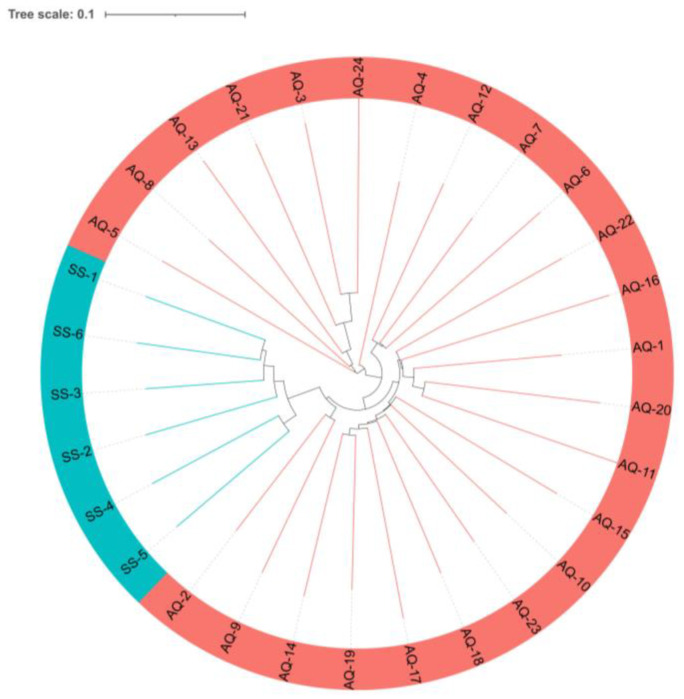
Evolutionary tree constructed from copy number variation regions (CNVRs); pink stripes and light blue stripes represent the AQ and SS groups, respectively.

**Figure 4 genes-14-00472-f004:**
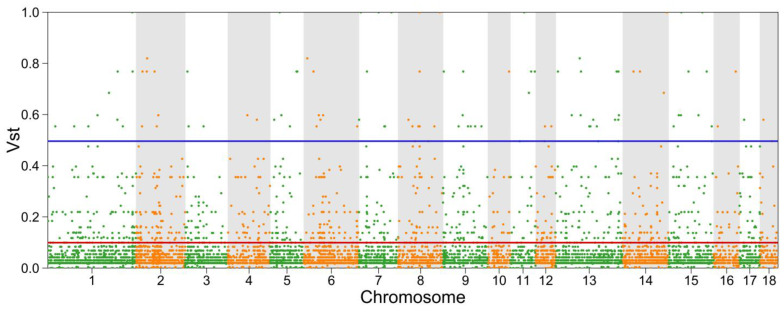
The V_ST_ values distribution of copy number variation regions (CNVRs) in AQ and SS pigs among autosomal chromosomes. The red line and blue line represent the top 10% (V_ST_ = 0.11) and top 1% (V_ST_ = 0.55) levels. Points located above the blue line were identified as selected CNVRs for AQ pigs.

**Figure 5 genes-14-00472-f005:**
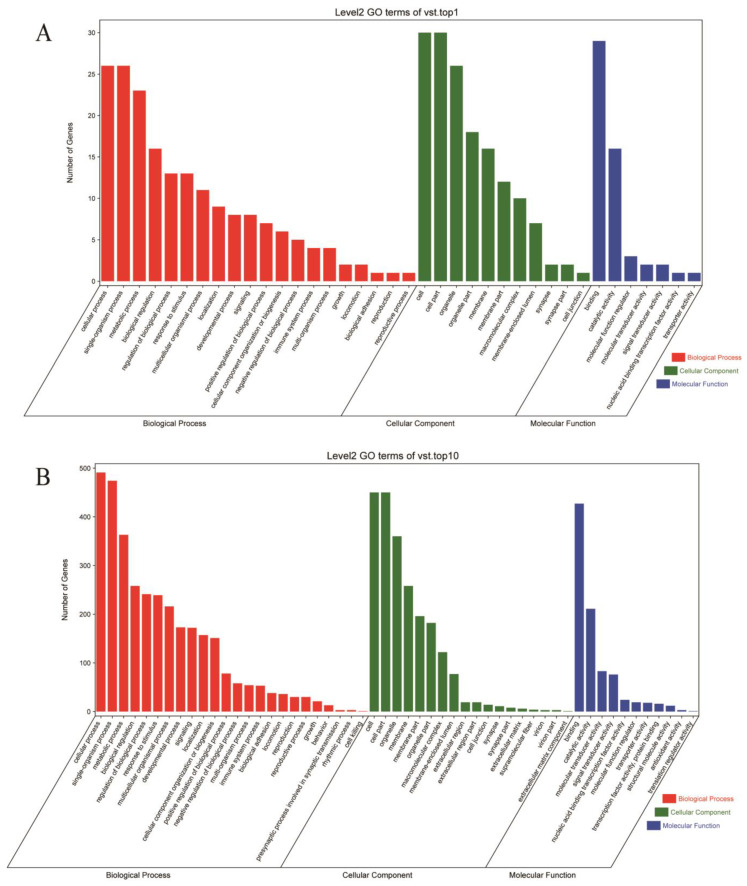
Gene ontology (GO) terms of enrichment of group-differential CNVRs. (**A**,**B**) represent GO terms of the top 1 and top 10 group-differential CNVRs, respectively; bars with different colors represent biological processes, cellular components, and molecular functions.

**Figure 6 genes-14-00472-f006:**
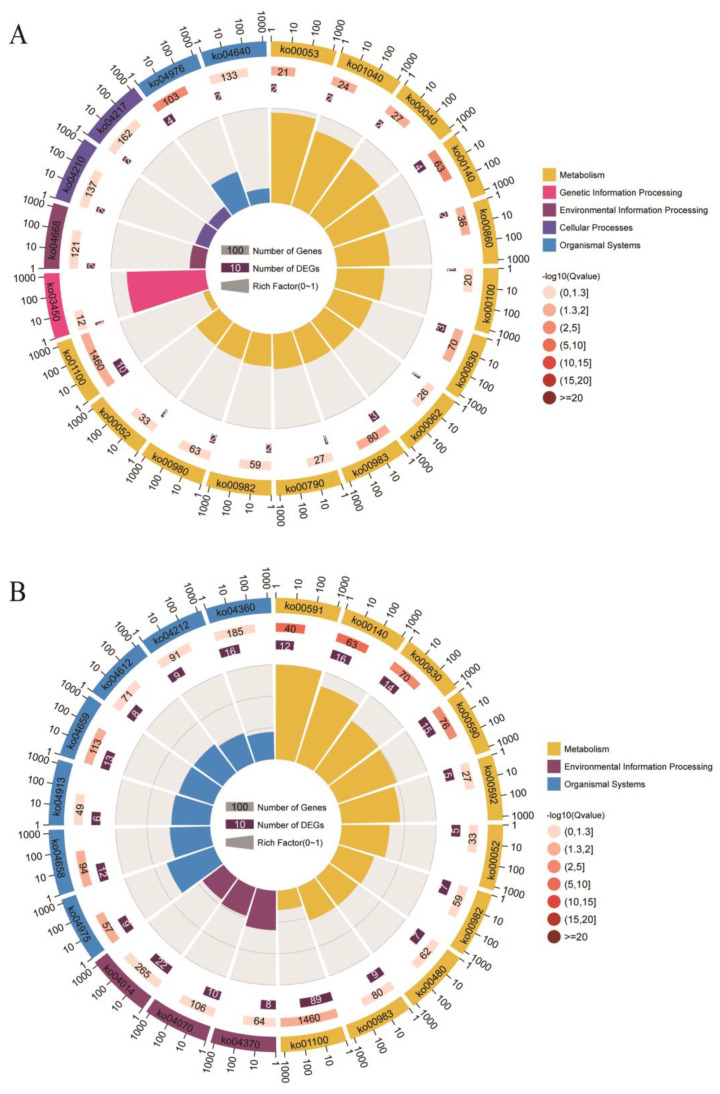
KEGG pathway enrichment of group−differential CNVRs. (**A**,**B**) represent KEGG pathways of the top 1 and top 10 group−differential CNVRs, respectively. The circles represent the pathway ID, number of background genes, and number of group−differential CNVRs overlapping genes from outside to inside. The number outside the circle represents the quantity co−ordinate. Length of bars represents the rich factor of the relative pathway.

**Table 1 genes-14-00472-t001:** The distribution of CNVRs in each chromosome of pig genome. Numbers of individual type and total CNVRs on 18 autosomal chromosomes were listed in different populations.

Chr	Length	All			AQ			SS		
		Total	DUP	DEL	Total	DUP	DEL	Total	DUP	DEL
1	274.33	1175	641	446	1157	639	431	457	117	287
2	151.94	1031	557	369	1016	556	357	445	147	225
3	132.85	607	316	243	601	315	238	227	47	153
4	130.91	574	276	249	566	276	242	261	78	154
5	104.53	587	292	235	581	292	230	258	70	155
6	170.84	865	475	308	856	473	304	328	93	195
7	121.84	561	274	208	548	272	199	257	66	136
8	138.97	569	226	302	558	222	296	251	47	186
9	139.51	652	320	252	642	320	244	295	81	175
10	69.36	290	109	176	286	108	174	153	31	118
11	79.17	283	74	198	278	74	193	144	18	120
12	61.6	394	210	140	387	209	136	158	42	95
13	208.34	874	445	361	861	445	348	363	102	226
14	141.76	672	351	259	668	350	257	276	84	154
15	140.41	504	262	217	492	261	208	206	56	132
16	79.94	287	81	204	283	81	200	157	22	133
17	63.49	328	132	167	324	132	163	165	38	108
18	55.98	176	70	100	175	70	99	78	8	65
Total	2265.77	10,429	5111	4434	10,279	5095	4319	4479	1147	2817

## Data Availability

Data are available from the National Center for Biotechnology Information (NCBI) database under the accession number PRJNA699491.

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
