# Peer review of "Copy Number Variation Analysis Revealed the Evolutionary Difference between Chinese Indigenous Pigs and Asian Wild Boars"

_genes, 2023, doi:10.3390/genes14020472_

Round 1
Reviewer 1 Report
The manuscript reports a simple descriptive study that reported copy number variations in 24 pigs of a local Chinese breed and in 6 Asian wild boars. The manuscript should be complely re-written, results re-interpreted and the design should be modified.
The title is completely misleading and not properly presented in English - there is no association study that is then presented in the text.
The abstract is also competely misleading - it is not clear what was done and what are the relevant results of this study - the flow of the study is puzzling and not correct and the reported conclusions are not derived by what was done.
Introduction is randomly organized - the first part from lines 26 to 42 is not relevant - it is not clear the meaning of the subsequent part. The aim of the study is not in line with the title and define just a descriptive work.
Materials and methods
It is not clear the difference between CNVs and CNVRs. Functional enrichment is not described properly - and it doe not make any sense in this context.
Compared Analysis of the CNVRs with QTL Database - it is not clear its meaning and it is not described to give the possibility to understand how it was done - and the relevance of the derived results.
Protein‒Protein Interaction (PPI) Network Analysis: not appropriate for this study - it is not clear its meaning and how it was done.
Results - the presentation is poor and the relevance of the results obtained it very limited. Legends of the figures and table are poor - there are some parts derived by the critical parts indicated in the Material and methods that should be eliminated. Discussion should be completely re-written.
English is very difficult to understand in several parts.
Reviewer 2 Report
What is group SS
Line 138-142 We found that the density of CNVRs on each chromosome and population varied. The average densities of both population, AQ and SS populations were 4.6, 4.54, and 1.98, respectively (It is not clear what the author wanted to say)
Line 164-165 To further discover the differential CNVRs between the two groups, we referred tothe parameter VST to filter all CNVRs (The parameter VST not presented in 2. Materials and Methods)
Figure 1. Poor quality. It is not clear where the AQ is shown, and where the AWB.
Figure 5, Figure 6 - Very poor quality, can't see anything.
The results obtained in the work are not obvious. Two groups were compared (n=30 and n=6), they are not balanced and the approach to assessing the differentiation between populations can give large errors. In addition, the analysis of the results relative to earlier results by other researchers has not been carried out and the contribution of this work to the general topic is not clear.
Round 2
Reviewer 1 Report
The manuscript has been modified after the first version. However, it is still not acceptable and further corrections are needed.
Abstract: it is still not informative: it must mention the method used to detect CNV (e.g. NGS), the number of animals sequenced and the depth of sequencing. In addition, it is not clear what is VST and the parameter used to call CNV.
Introduction: from lines 34 to 42 - the text is not relevant for this study and should be removed.
Line 43: Sus scrofa, also known as pigs, have a long history of over 40 million years - it does not make any sense. Please cancel.
"To better understand the evolution of meat quality and other traits, high-throughput sequencing was used in this study ..."
From just sequencing data it is not possible to obtain the stated information. Please rephrase.
Line 50: Here,the definition of the acronym AQ is not correct.
Lines 51-72: It does not mention any previous studies in pigs that investigated CNV and the use of this information derived from Whole genome sequencing. This part must be improved substantially
Materials and methods:
Lines 93-94: The average depth of these data was ten generations. No, this sentence is not correct - ten generation? of what?
Paragraph 2.2. What is the meaning of CNV selection ?
Paragraph 2.3. What is CNVR selection?
This part is not clear at all: What represent VST ? It is not clear if it is appropriate its use in this context.
Results
3.1. What are DUP and DEL?
The provided coverage of CNV on the pig genome is quite worrying - this fraction is not in line to what reported in previous studies aaand in other species.
What is the differnt between high confidence CNVR and other CNVR?
Lines 149 - ... the authors stated that there was no proportion between chromosome length and distribution of CNVR. This is not clear: from what was reported, it seems that there is a relationship between these two parameters. However, this should be test with a correlation based on the chromosome size, CNVR number and size.
Paragraph 3.3. It is completely not explained and it meaning quite puzzling - It should be substantially revised, starting from M&M
Paragraph 3.4. The content is not very relevant in this context. It would make sense only if there would be differences between the two groups of investigated pigs.
Discussion
It should be substantially revised - according to the limits and problems evidenced in Results. Only relevant differences between the two groups of pigs should be discussed.
Conclusions.
The last sentence should be deleted.
Reviewer 2 Report
Accept in present form
Author Response
Thanks for the positive comments about our manuscript and your acceptance. You were very responsible in reviewing articles. Your advices were very helpful for revising and improving our paper, as well as the important guiding significance to our researches.